# An Assessment of Agricultural Vulnerability in the Context of Global Climate Change: A Case Study in Ha Tinh Province, Vietnam

**Duong Thi Loi** [1,*]**, Le Van Huong** [2,*]**, Pham Anh Tuan** [3]**, Nguyen Thi Hong Nhung** [3]**, Tong Thi Quynh Huong** [3] **and Bui Thi Hoa Man** [3]

1   Faculty of Geography, Hanoi National University of Education, Hanoi 100000, Vietnam
2   Institute of Geography, Vietnam Academy of Science and Technology, Hanoi 100000, Vietnam
3   Faculty of Social Science, Tay Bac University, Son La 34100, Vietnam; phamtuan@utb.edu.vn (P.A.T.); nhungnguyen@utb.edu.vn (N.T.H.N.); tqhuong87@utb.edu.vn (T.T.Q.H.); buihoaman@utb.edu.vn (B.T.H.M.)
*   Correspondence: loidt@hnue.edu.vn (D.T.L.); lvhuong@ig.vast.vn (L.V.H.)

**Abstract:** Climate change is considered a major challenge of mankind in the 21st century. Agriculture is known as one of the most vulnerable sectors to climate change. This study aims to evaluate agricultural vulnerability to climate change in Ha Tinh province. The agricultural vulnerability zoning map is presented by using the index method with eight influential indicators across three components: exposure, sensitivity, and adaptive capacity. Agricultural vulnerability is quantified and classified into five levels, namely very low, low, moderate, high, and very high. The results indicated that Ha Tinh's agriculture was heavily affected by climate change, and the most vulnerable area is found in Huong Khe district and Thach Ha district. People and their activities play an important role in mitigating the vulnerability of agriculture to climate change. The results obtained from this study provide useful information for planning and determining guidelines to help farmers in the area.

**Keywords:** agricultural vulnerability; climate change; Ha Tinh province; vulnerability assessment; agricultural vulnerability index

## 1. Introduction

Climate change is one of the biggest challenges threatening regional and global security and undermining much that people have achieved in both the present and the future. It has a negative impact on people for socio-economic reasons, and also has a negative impact on ecosystems [1]. Due to a tight dependence on climate and weather conditions, agriculture is one of the most vulnerable sectors to climate change [2–4]. Climate change affects agriculture in multiple ways, such as through variations in the global average temperature and rainfall and extreme weather patterns [5]. Floods, droughts and other natural disasters continuously occur in many regions and territories around the world and, as such, threaten the livelihoods of farmers who depend upon agriculture [6]. Climate change may lead to crop failure in many parts of the world and humanity will face possible food security challenges and political instability [7]. Raw material shortages delay the production within some industries that depend on agricultural raw materials, thereby threatening national economies and the global supply chains within these respective industries [8]. The Intergovernmental Panel on Climate Change [9], the leading international scientific body for climate change assessment stated "Vietnam is one of the countries likely to be most affected by climate change due to its extensive coastline, vast deltas, and floodplains, location on the path of typhoons". According to the report of the Ministry of Agriculture and Rural Development [10], Vietnam may lose 50% of the area currently used to cultivate rice due to rising sea levels, seriously threatening the food security of millions of people. Furthermore, the average damage caused by climate change to agriculture and rural areas

is estimated to be 36 million USD. In 2030, the yield of rice, maize, and soybeans in Vietnam may be reduced by 8.37%, 18.71%, and 3.51%, respectively [11]. Therefore, the study of agriculture vulnerability assessment due to climate change is an urgent problem for an agricultural country like Vietnam.

Vulnerability is the degree to which a system is susceptible to or unable to cope with the adverse effects of climate change including climate variability and extremes [12]. Agricultural challenges in the context of climate change were presented in much of the literature with different spatial scales. Vulnerability in agriculture is defined as the degree to which an agricultural system is susceptible to or unable to cope with the negative impacts of climate change, including climate change and extreme weather events [13]. Agricultural vulnerability (AV) is studied in general from the global level [2,14–17] to the regional level [18–21] and national or local level [22–26]. The assessment of AV to climate change may be conducted by many methods. One of them is addressed by statistical analysis. For example, based on climate statistical data such as precipitation, evapotranspiration, surface water availability, depth to groundwater, etc., and yield from agriculture, the variation in climate and yield declines were analyzed to clarify the AV [27,28]. The data used in this method is often secondary data, so it is cheap and is less time-consuming. Besides, the data can be used and re-used to check different variables. However, the disadvantage of this method is that it is difficult to consider the adapted capacity and secondary data can be easily misinterpreted. The second method is model simulation. Most of the modes were used to focus on the group of factors most sensitive to climate change in the study area [5]. Some of the models have been applied widely in the field of agricultural management such as the Agricultural Catchments Research Unit (ACRU) model [29], Decision Support System for Agro-technology Transfer model (DSSAT), the Soil Water Balance (SWB) model [30], Environmental Policy Integrated Climate (EPIC), the (Agricultural Production Systems Simulator) APSIM [31], and AquaCrop [32]. The model allows for assessing crop responses under different climate change scenarios. Although the use of the model simulation is very effective in evaluating AV, it requires relatively large and detailed data. For developing countries, the assessment of AV by this method is difficult due to the lack of a database. Existing data were often available only on the country level or in very select places [33].

A common method used to quantify the vulnerability to climate change is the index method [34–36]. In this method, the selection of appropriate indicators used to assess vulnerability to climate change is very important, it closely depends on the research objective and study area [37]. In this case, agriculture is considered to be the objective of the study. Normally, agriculture or the growth of plants and animals is often influenced simultaneously by two groups, namely natural factors and socio-economic factors. From that, natural factors (e.g., temperature, rainfall, extreme weather, etc) and socio-economic factors (e.g., labor level, agricultural income from crop production, irrigation, etc) will be considered as input indicators [38,39]. It can be said that the index method brings a comprehensive approach and is considered suitable for assessing vulnerability as vulnerability is a theoretical phenomenon that cannot be measured as an observed phenomenon [40–42]. According to IPCC, AV is the function of three components of vulnerability to climate change including exposure, susceptibility, and adaptive capacity. IPCC has taken the definition of these components. Accordingly, exposure (E) refers to the extent and the characteristics of a system exposed to significant climate variability. Sensitivity (S) means the degree of influence as a system stimulated by climate-related factors. Adaptive capacity (AC) is understood as the ability to make a profit and avoid loss as the natural and man-made systems are affected by the negative impact of climate change. The relevant natural or socio-economic indicators are classified into the above three groups of components. Although this method proved to be very effective and applied to many parts of the world, there have not been enough thoroughly researched projects on AV conducted in Vietnam so far. In this context, the objectives of this study are: (1) Determine and assess the impact of indicators on agricultural vulnerability; (2) Classification of agricultural vulnerability in the study area to climate change. As such, three components were captured via eight indicators:



(1) Variation coefficient of precipitation; (2) Heavy rain; (3) Hot days; (4) Risk of flash flood; (5) Percentage of agricultural land; (6) Percentage of inundated area; (7) Household below poverty line; and (8) The density of irrigation works. Values of indicators were normalized and a relationship matrix among indicators was built in the GIS environment. As a result, the level of AV in the study area is classified into five levels: very low, low, moderate, high, and very high based on the range of AV. The results obtained from this study provide useful information for planning and determining guidelines to help farmers in the area.

## 2. Study Area

Ha Tinh is one of six provinces situated in the North Central coastal provinces with a total area of 6026 sq. km. It is located between 17°54′–18°38′ N latitude, 105°11′–106°36′ E longitude. Ha Tinh is bordered by Nghe An to the North, Quang Binh to the South, Lao People's Democratic Republic to the West, and the East Sea to the East, with more than 137 km of the seashore. It is subdivided into 13 district-level subdivisions (Figure 1).

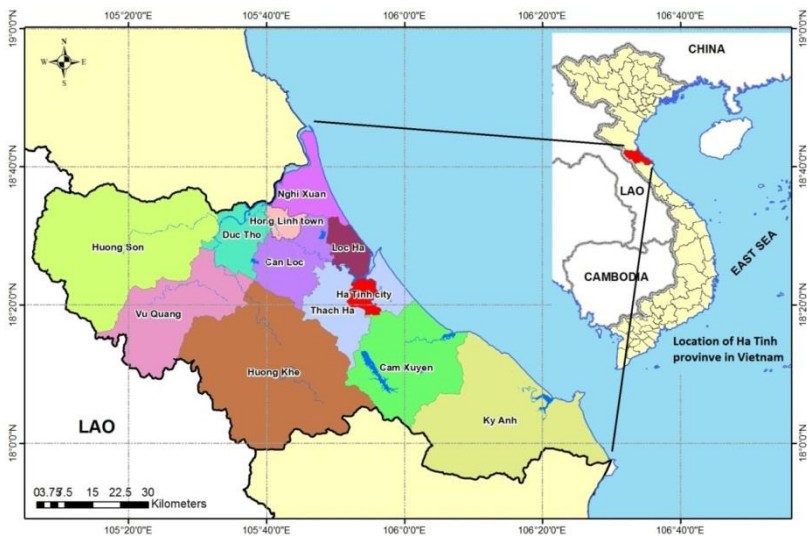

**Figure 1.** Location of study area.

Ha Tinh's terrain is both narrow and sloping, tilting towards the east. The mountain ranges are located to the west with an average height of 1500 m, followed by a range of hills, small and narrow elongated plain, and coastal sandy banks. The mountainous terrain accounts for 80% of the province's natural area that is sharply differentiated and divided, forming different ecological areas.

Ha Tinh is located in the tropical monsoon climate area with an average annual temperature of 23.7 °C. The annually average rainfall is over 1500 mm and is concentrated mainly in the rainy season from August to November. The study area is often affected by natural disasters (e.g., storms, flash floods, erosion) and extreme weather phenomena (e.g., frost, Foehn winds) [43].

## 3. Material and Methods

### 3.1. Identify Vulnerability Indicators for Calculating AV

Data was collected from diverse sources. In which, climate-related indicators such as temperature and precipitation were extracted from the climate statistical yearbook, then analyzed and calculated by the authors. These are secondary data and district-level data. The collected data is from 1980 to 2013. The weather-related data was initially measured directly from the meteorological observation stations. There are four meteorological observation stations in Ha Tinh province, namely Kim Chuong, Huong Khe, Ha Tinh, and Ky Anh. After that, these data are processed and recorded in the district's climate data

report. Data from local stations are sent to the provincial hydrometeorological center for monitoring, research, and disaster management.

Table 1 describes indicators and their source. All spatial analysis processes were conducted in ArcGIS, version 10.4, software in the WGS84 projection. The exposure map was the combined result of three main indicators, namely variation coefficient of precipitation, heavy rain, and hot days. Data on indicators of sensitivity and adaptive capacity were collected from the following sources: Statistical yearbook of 12 district-level administrative units in Ha Tinh province; report on the implementation of socio-economic tasks of the province and districts in Ha Tinh province; and the official website of Ha Tinh province.

**Table 1.** Components and indicators for calculating the AVI.

| Component | Indicators | Description | Impact on AV | Data Source |
|---|---|---|---|---|
| | Variation coefficient (Vc) of precipitation | The data used for the assessment was the Vc of precipitation from 1980 to 2013, which was taken from the climate data report of Ha Tinh province. Vc was interpolated using Inverse Distance Weighting method with a resolution of 0.00382 degrees, then was normalized to be valued from 0 to 1. | Rainfall is an integral part of food production. It is estimated that up to 60% of staple food is produced from rain-fed agriculture [44,45]. Irregular rain patterns accompanied by storms and tropical depressions cause crop failure and severe flooding. | [46] |
| **I. E** | Heavy rain | A day is called a "heavy rainfall day" according to Vietnam Meteorological Department if the rainfall is above 50 mm/day. The data used is the average number of heavy rainy days in the period of 1980—2013, which was taken from the climate data report of Ha Tinh province. The number of heavy rain from observation stations was also interpolated using Inverse Distance Weighting method with a resolution of 0.00382 degrees, then was normalized to be valued from 0 to 1. | Too much rain can harm crop production, floods fields, and wash away seeds and precious topsoil. Wet weather encourages bacteria and fungus growth, which can further damage crops [47]. | [46] |
| | Hot days | The "hot day" is defined as days with a temperature above 35 degrees Celsius. The data used is the average number of hot days in the period of 1980–2013. The data was processed by using the same weather station and method as precipitation data. | Temperature plays a very important role in the development of organisms. The increased temperature would affect the crop calendar in tropical regions. Global warming can reduce the length of the effective growing season, particularly where more than one crop per year is grown, and reduce global yields [48]. For example, each degree-Celsius increase in global mean temperature would, on average, reduce global yields of wheat by 6.0%, rice by 3.2%, maize by 7.4%, and soybean by 3.1% [49]. | [46] |

**Table 1.** *Cont.*

| Component | Indicators | Description | Impact on AV | Data Source |
|---|---|---|---|---|
| **II. S** | Risk of flash flood | The assessment is based on the flash flood risk zoning map. Risk of flash flood data is stored in shapefile format in ArcGIS software. The risk of flash floods in the study area is divided into four levels: high risk, medium risk, low risk, and no risk. | An increase in flash floods in terms of quantity, intensity, and frequency is seen as a manifestation of climate change. The flash flood has caused many negative effects on the environment and society [50] and is the top weather-related killer. The flash flood hazard map is built based on a combination of physical factors such as slope, land use-land cover, soil type.etc. The areas at higher risk of flash floods have greater vulnerability of agriculture, and vice versa. | [51] |
| | Percentage of agricultural land | This indicator is determined by the ratio between the area of agricultural land and total natural area by each district. Percentage of agricultural land data is in excel file format, stored into a shapefile in ArcGIS software. Finally, the percentage of agricultural land was classified into five classes. | The proportion of agricultural land is an important factor in assessing the impact of agriculture on climate change. Accordingly, areas with a large proportion of agricultural land area have a higher index of agricultural vulnerability due to climate change than the rest of the region. | [46] |
| | Percentage of inundated area | This indicator is determined by the ratio between the inundated area and agricultural land by each district. Percentage of agricultural land data are also in excel file format, stored into a shapefile. It was further classified into five classes. | Agricultural production activities in the study area are concentrated mainly in lowland and coastal areas. However, these are also areas that are more susceptible to floods. In areas with large inundated areas, the vulnerability to climate change is considered to be higher than the rest. | [52] |
| **III. AC** | Household below poverty line | This indicator is identified by the percentage of households below poverty line to the total population. From 2016 to now, Vietnam has applied a national multidimensional poverty measure, based on the Alkire–Foster method with five elements: (1) living conditions; (2) income level; (3) access to health and education; (4) access to information; and (5) access to security insurance and social assistance [53]. Number of households below poverty line is converted to attribute data in ArcGIS software and stored into a shapefile. It was further classified into five classes. | The poverty rate is one of the key factors in the vulnerability of households and communities to climate change, and their adaptive capacity. Areas with high poverty rates have a very low response to climate change. | [54] |

**Table 1.** *Cont.*

| Component | Indicators | Description | Impact on AV | Data Source |
|---|---|---|---|---|
| | The density of irrigation works | The index is determined based on the number of irrigation works per unit area of agricultural land (hectare). This data is collected from the Statistical Yearbook of Hatinh province and then processed using the same method as household below property line indicator. | Irrigation works play an important role in agriculture. They help to regulate water sources in agricultural production, minimizing flooding in the rainy season and drought in the dry season, thereby increasing resilience to climate change | [46] |

*3.2. Calculation of the Agricultural Vulnerability*

After determining the indicators in each component, the data is processed in GIS environment. In the next step, the value from maps is normalized. The purpose of the normalization procedure is to adjust different factors with different units to the same dimensionless unit. As a result, the values of the indicators range from 0.0 to 1.0. The value at 0 represents the least impact and the value at 1 represents the greatest effect. Normalization of indicators is implemented by using the following formula [55].

$$X_{ij} = \frac{X_{ij}(t) - Min X_{ij}}{Max X_{ij} - Min X_{ij}} \tag{1}$$

where, $X_{ij}$ the normalized value of $j$ vulnerability indicator in the $i$th object, $X_{ij}$ $(t)$ is the real value of $ij$th vulnerability indicator, $Min$ $X_{ij}$: the minimum real value of the $ij(t)$ vulnerability indicator in all objects, $Max$ $X_{ij}$: the maximum real value of the $ij(t)$ vulnerability indicator in all objects.

Finally, the AV is calculated by the equation below [55]

$$AV = 1/3 \ (E + S + 1 - AC) \tag{2}$$

where, AV: Agricultural Vulnerability, E: Exposure, S: Sensitivity, AC: Adaptive Capacity. The methodology is illustrated in the flowchart, as shown in Figure 2.

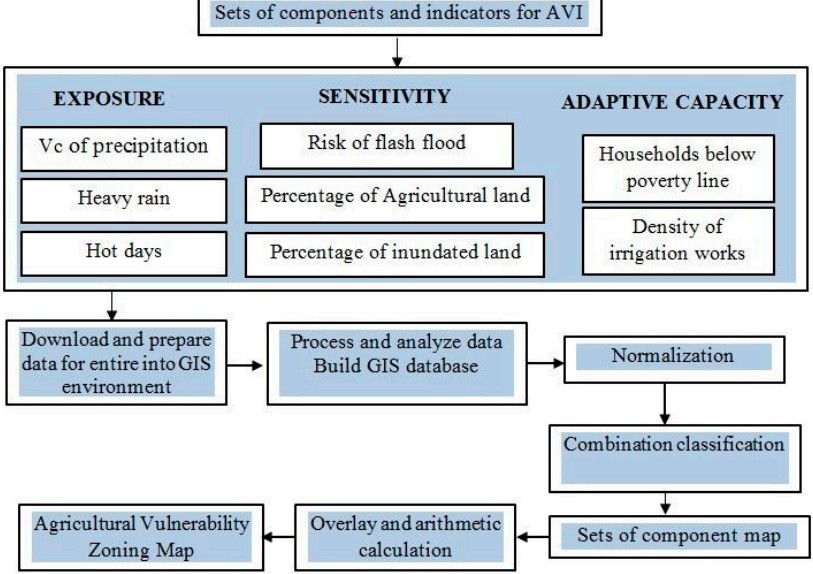

**Figure 2.** Methodology flowchart for AVI.

## 4. Results

*4.1. The Component Analysis Results for the Study*

(1)    Exposure component

As described in the research methods section, the E map was built based on three indicators, namely Cv of precipitation, heavy rain, and hot days. The pixel size of interpolation maps is assigned at 0.00382. As a result, Cv of precipitation was scaled from 19.002% to 25.9998%, in which the erratic rainfall area was recorded in the middle east of Ha Tinh province with a variable frequency of nearly 26%. In contrast, in the southeast area the rainfall was more stable (Figure 3a). The second indicator is the number of heavy rain days. Its value ranged from 9.8002 to 14.0999 days. The area with frequent heavy rain was found in the eastern coastal plain (Figure 3b). The number of hot days is the third important indicator to assess the E. There was a significant difference in the number of hot days in a year among regions. In particular, the west and southwest areas had a very large number of hot days in a year; up to nearly 70 days/year. The number of hot days in the eastern part was significantly less than in the west, ranging from 40 to 50 days (Figure 3c).

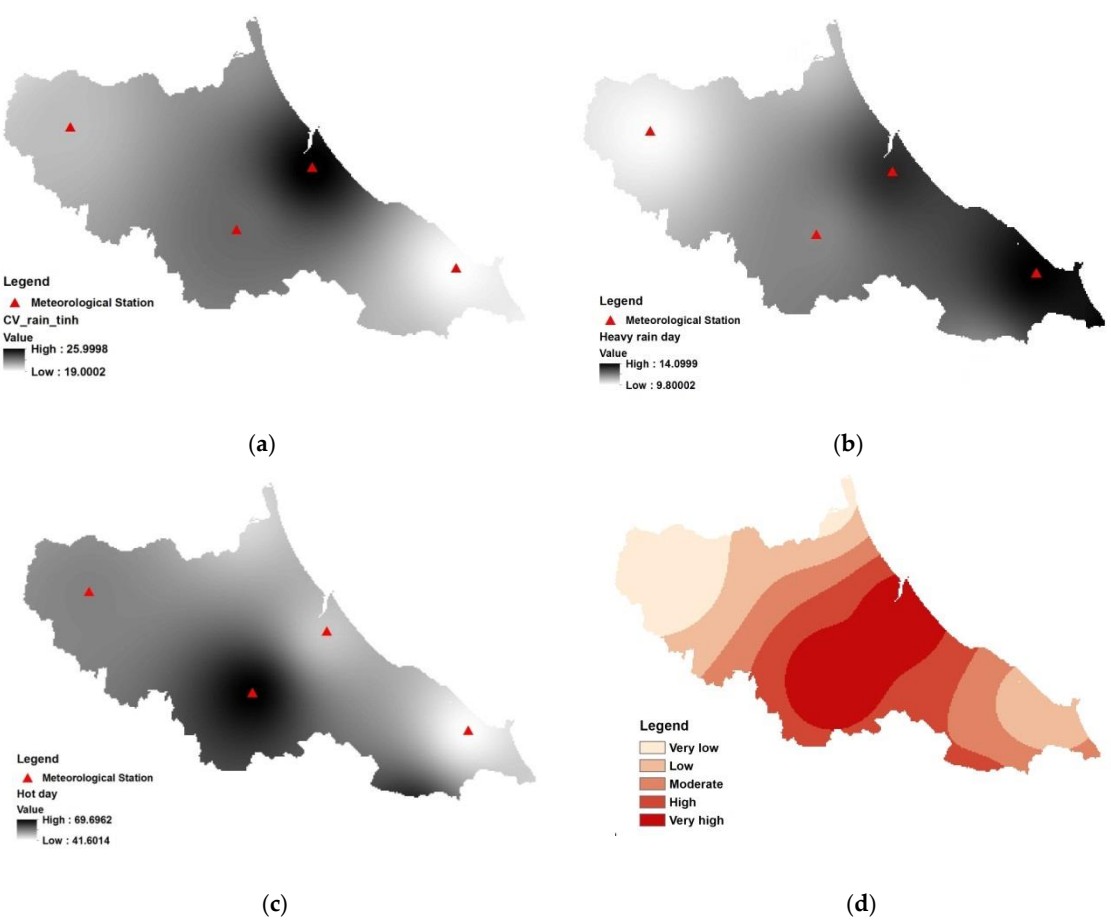

**Figure 3.** Indicators for creating an exposure component: (**a**) Cv of precipitation indicator; (**b**) Heavy rain days indicator; (**c**) Hot days; (**d**) Exposure component.

The values of the indicators were normalized to the same units, and then overlaid to build the E map. As a result, the values of E were scaled from 0.215 to 0.682. The E map was classified into five levels; very low, low, moderate, high, and very high, with the corresponding values as shown in Table 2. The most exposure was distributed in a strip in the middle and the center of the study area, within the provinces of Huong Khe, Thach Ha, and Ha Tinh city. Areas north and south had lower exposure levels (Figure 3d). This is also

reasonable as the high-risk area is located in the path of storms and tropical depressions in central Vietnam.

**Table 2.** Classification of components and AVI.

| Components | Scale Classification | Rating |
|---|---|---|
| E | Less than 0.3 | Very low |
| | From 0.3 to 0.4 | Low |
| | From 0.4 to 0.5 | Moderate |
| | From 0.5 to 0.6 | High |
| | Greater than 0.6 | Very high |
| S | Less than 0.25 | Very low |
| | From 0.25 to 0.4 | Low |
| | From 0.4 to 0.55 | Moderate |
| | From 0.55 to 0.7 | High |
| | Greater than 0.7 | Very high |
| AC | Less than 0.2 | Very low |
| | From 0.2 to 0.4 | Low |
| | From 0.4 to 0.6 | Moderate |
| | From 0.6 to 0.8 | High |
| | Greater than 0.8 | Very high |
| AV | From 0.32–0.41 | Very low |
| | From 0.41–0.47 | Low |
| | From 0.47–0.53 | Moderate |
| | From 0.53–0.6 | High |
| | From 0.6–0.67 | Very high |

(2)    Sensitivity component

From the above three indicators, S map was created. The flash flood risk zoning map was referenced from the research work of the Ministry of Agriculture and Rural Development in Ha Tinh province [51]. Accordingly, the map was established based on an integrated assessment of factors affecting flash flood risks such as slope, land use/land cover, soil texture, and drainage density. The flash flood risk was classified into four levels: very low, low, moderate, and high (Figure 4a). The second indicator to assess the S component is the percentage of agricultural land. Figure 4b showed that agricultural land occupies a large proportion in the eastern part. This is a coastal plain that is favorable for growing a paddy and other crops. Districts with a large proportion of agricultural land were recorded in Duc Tho, Can Loc, Loc Ha, of which the largest proportion of agricultural land is found in Duc Tho district with 53%. In the western part, the percentage of agricultural land is less than 12% as this area is covered mainly by high mountains and natural forest. Figure 4c described visually the context of inundation levels through the percentage of inundated areas in the study area. As a result, inundated areas were mainly concentrated in the eastern part, typically Ha Tinh city over 12.7%, and Loc Ha 11.4%. The main reason was due to the storms originating from the East Sea. They often cause heavy rain and serious floods in the eastern coastal section of the study area. In the west part, the influence of the storm was weakened, so the rain also decreased slightly and caused less flooding.

As presented in the method section, the S map is the combined result of the three indicators mentioned above. As a result, the value of E was scaled from 0.2 to 0.8 and is classified into five classes, including very low, low, moderate, high, and very high. Table 2 described the value corresponding to each level. Accordingly, the values were less than 0.25 corresponding with very low level. Similarly, low level, moderate level, high level, and very high level were given the value from 0.25 to four, from four to 0.55, from 0.55 to seven, and greater than seven. Figure 4d showed that the northeast region was more sensitive to climate change than the rest. The districts with high sensitivity are found in Duc Tho, Ha Tinh, Loc Ha, and Can Loc with E values greater than 0.55.

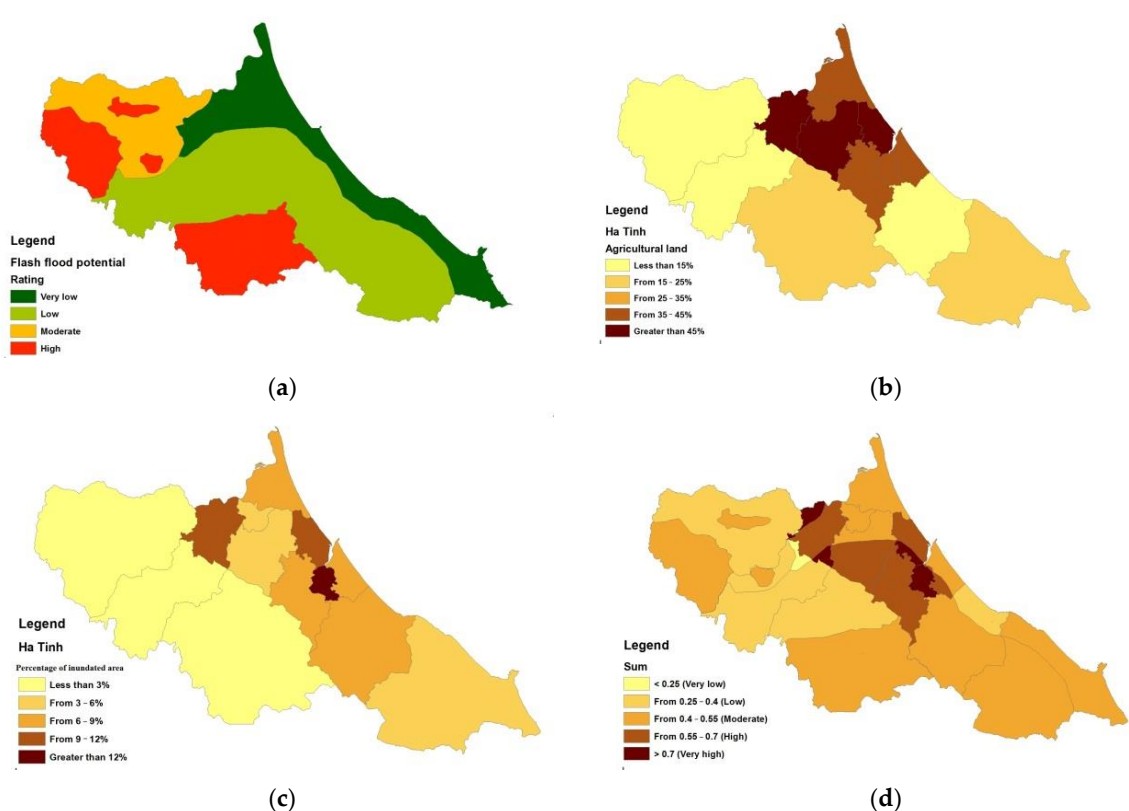

**Figure 4.** Indicators for creating a sensitivity component: (**a**) Flash flood risk area; (**b**) Percentage of Agricultural land; (**c**) Percentage of inundated area; (**d**) Sensitivity component.

(3)    Adaptive Capacity

AC component was assessed based on two indicators, namely households below the poverty line and density of irrigation works. It can be said that poverty is one of the key factors in the vulnerability of households and communities to climate change, and their adaptive capacity. According to statistics of Ha Tinh Department of Labor, War Invalids and Social Affairs [56], although the poverty rate in Ha Tinh has decreased rapidly from 23.1% (2010) to 3.51% (2020), Ha Tinh is still ranked as one of the provinces with the highest poverty rate in Vietnam. The research result showed that poverty rates are higher in the districts in the west than in the districts in the east. The harsh physical conditions have caused great difficulties for the socio-economic development of the districts in the west. The west is a rugged mountainous area, which is often affected by natural disasters such as flash floods and unusual colds, landslides, and Foehn wind. In addition, the west is the main living area of ethnic minorities, with low education levels. Livelihoods closely depend on natural resources. The two districts with the highest poverty rate are Huong Khe and Vu Quang with values of 5.47% and 5.38%, respectively. The lowest poverty rate was assigned Ha Tinh city with 1.32% (Figure 5a). Areas with a high percentage of poor households are less able to adapt to climate change.

Infrastructure plays a strategic role in creating a larger multiplier effect in the economy with agricultural growth [57]. Irrigation is a very important part of the agricultural infrastructure system. It contributes to the regulation of water resources—one of the most important factors of agricultural production. Therefore, irrigation is selected as an indicator to assess the adaptability of agriculture to climate change. As the result, Cam Xuyen had the largest number of projects with more than 1.2 works/hectare of agricultural land while Vu Quang and Huong Khe had the least density of irrigation works (less than 0.1 works/hectare) (Figure 5b).

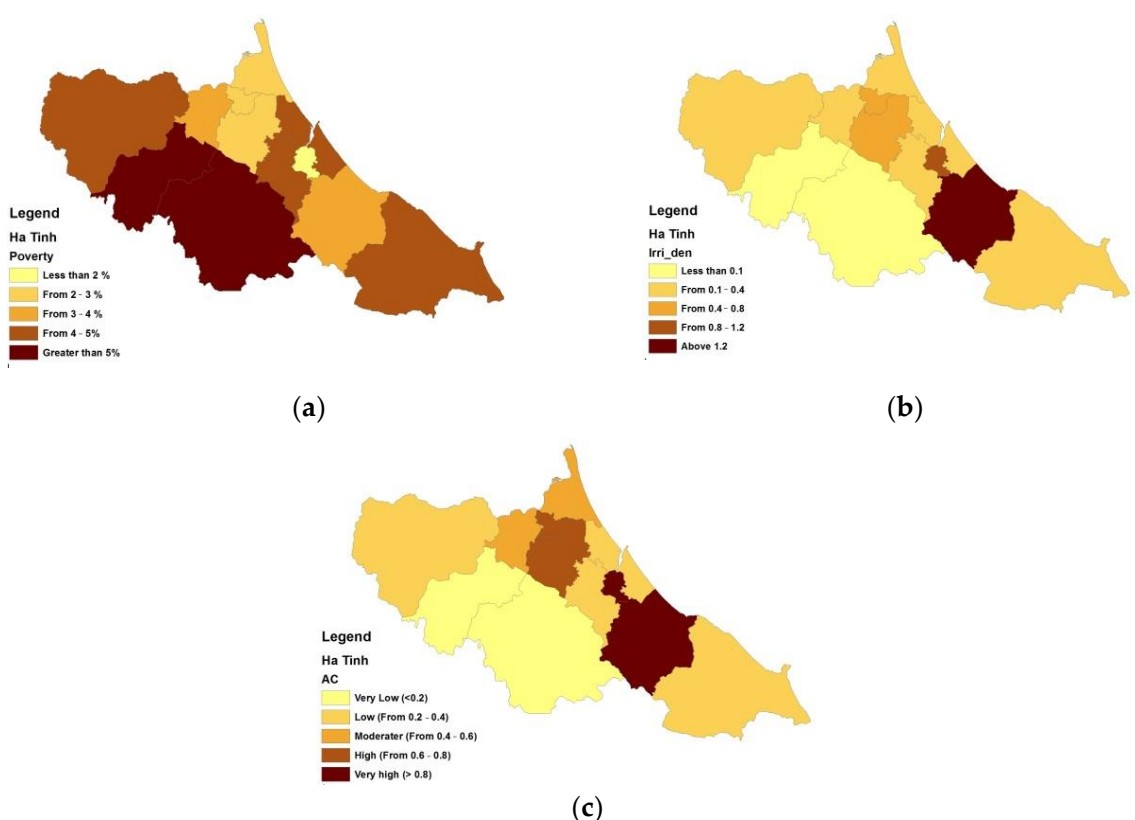

**Figure 5.** Indicators for creating an Adaptive Capacity component: (**a**) Household below poverty line; (**b**) Density of irrigation works; (**c**) Adaptive Capacity component.

Figure 5c clearly shows that high adaptive capacity mainly appeared in urban areas, cities, or towns. Low AC was found mainly in mountainous districts. Hong Linh town and Ha Tinh city are the two places with the best ability to adapt to climate change whereas mountainous districts (Vu Quang, Huong Khe, Huong Son) have low adaptability to climate change. This difference is due to the fact that city and town areas are more favorable for disaster prevention than mountainous areas. In Ha Tinh, most of the mountainous districts have limited infrastructure for agriculture, there is almost no disaster warning station, so the damage in agriculture during a flood is huge. Besides, education levels are still low and the population consists mainly of ethnic minorities. Farming techniques in agriculture are outdated as they are mainly based on manual labor, so productivity is not high. Although in recent years the local government has implemented many solutions to bring science, technology, and modern machinery into agricultural production. However, due to the shortage of qualified workers, the implementation faces many difficulties [43].

### 4.2. Agricultural Vulnerability Zoning

As shown in Figure 6, AV of Ha Tinh province was scaled from 0.32 to 0.67 and classified into five classes, namely very low, low, moderate, high, and very high. Within these, a very low AV level ranging from 0.32 to 0.41 was recorded. Similarly, low AV, moderate AVI, high AV, and very high AV levels were found with values from 0.41 to 0.47, from 0.47 to 0.53, from 0.53 to 0.6, and from 0.6 to 0.67, respectively. The result showed that the agriculture of Ha Tinh was facing a high risk of vulnerability due to climate change. Areas with very high AV accounted for 22% of the total area. Areas with high, medium, low, and very low AV levels accounted for 25%, 13%, 33%, and 7%, respectively.

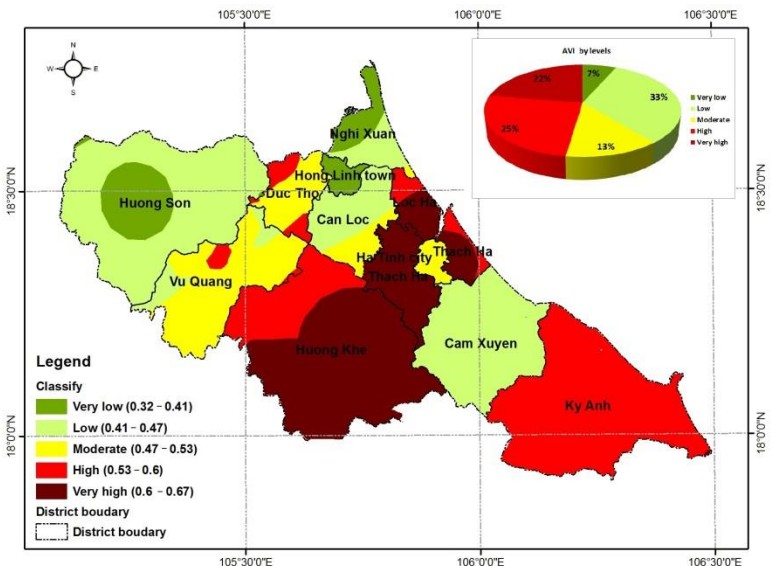

**Figure 6.** Agricultural Vulnerability zoning map.

The area with the highest AV level was observed in Huong Khe (93,272.9 ha), Thach Ha (31,756.1 ha), and a part of Loc Ha district (7939.8 ha). Areas with high agricultural vulnerability levels were found in Ky Anh (103,058.7 ha), Huong Khe (31,848.2 ha), and Duc Tho (5794.3 ha) (Figure 7). These areas are frequently affected by natural disasters such as floods, storms, and tropical depressions during the rainy season. In addition, these are all poor districts. Agriculture is the main production sector and keeps a high proportion in the economic structure [43], so the E index is quite high whereas the AC index of these districts is quite low. Areas with the least agricultural vulnerability were found in Hong Linh town, north of Nghi Xuan, and center of Huong Son district with an area of 5938.7 ha, 9901.5 ha, and 23,132.8 ha, respectively (Figure 7). These are areas with good irrigation systems for agriculture, so they are more active in agricultural production. Hong Linh town and Huong Son have a small area of agricultural land. Hong Linh town's economy mainly consists of industrial and service activities, while Huong Son's area is covered by natural forests. In addition, this is also an area with relatively stable weather conditions compared to the rest.

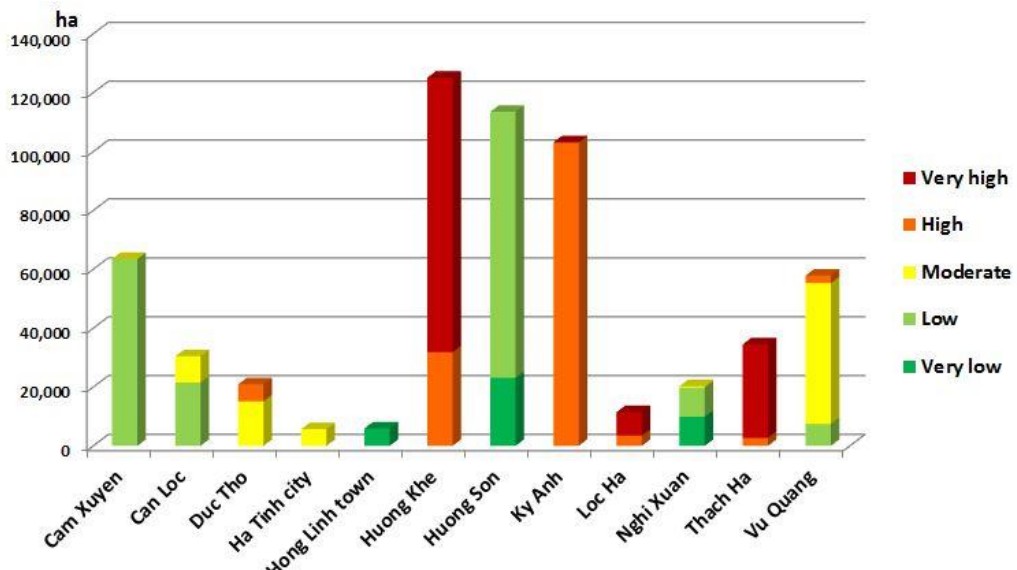

**Figure 7.** Classification of Agricultural vulnerability by districts.

## 5. Discussion

The index method often combines quantitative and qualitative assessment. This method has many advantages such as scientific objectivity, high reliability, and quick analysis, so it has been applied in many previous studies. However, this method also has certain limitations such as subjective factors of surveyors in the construction of any index; Contextual errors can affect survey content [13,37]. The present study also faces the same challenges as previous studies. To partially overcome the limitations caused by this method for previous works, the authors considered the following issues: (1) the selection of indicators should be made carefully and considered to be the most important task in this study. The selected indicators were based on the available literature and the actual context of the study area, combined with expert consultation. The survey was conducted by the authors to collect data and verify the result; (2) the proposed weights for the indices. In this study, eight indicators of three components were considered equally important for agricultural vulnerability to climate change. Therefore, the weights of all indicators were assumed to be equal. This helps to limit errors caused by human subjective factors.

The results of this study also address a challenge when using the regional quantitative assessments of vulnerability, that is, how to adjust the indicators to a particular context and function. In previous studies, much of the discussion focused on index selection and the conflict between desirable explanatory power and data constraints. However, such discussions did not specifically focus on variable thresholds and how significant the difference was concerning what the indicator is supposed to represent [26,57]. In this study, all components (E, S, AC, and AV) were classified into five classes, namely very low, low, moderate, high, and very high. Threshold intervals are different for each component or indicator, and they were adjusted to best suit a particular context and function.

This research, however, is subject to several limitations. Firstly, the data classification according to spatial boundaries is not uniform. While data related to climate (e.g., temperature, precipitation) or natural phenomena (flash flood) is a continuous variable, which is not limited to administrative boundaries, socio-economic data (e.g., percentage of agricultural land, percentage of the inundated area, household below poverty line) is a discrete variable and is counted by district-level administrative units. Although they were later normalized to return to the same unit of measure, at the boundary among the overlapping data layers, the accuracy was significantly reduced. This was found in Figures 4d and 6. Secondly, although all input indicators in the study are considered to be equally important in agricultural vulnerability assessment, this will partly eliminate the impacts caused by human subjective factor as it loses the distinction of the case study. There is no clear difference in the vulnerability of agriculture to climate change with other sectors. Moreover, the combination of quantitative data and qualitative data through discussions will enhance analysis and deepen the understanding of the problem. Thirdly, the database is hampered by limited access. Field surveys to collect information were carried out, however, the authors had to face many obstacles (COVID-19 epidemic, rugged mountainous terrain, poor transportation, illiteracy of ethnic minorities, etc.) Hence, the number of questionnaires collected was not reliable enough, and it was not included in this study. Many missing indicators have not been applied in this study.

On the one hand, and despite the above mentioned limitations, this study contributes to the debate on how to address the threat of climate change at a local level. It brings one more lesson learned about how to respond to climate change at the regional, national and global levels. Although research in this field has been mentioned in many previous studies around the world, it is the first in-depth study related to AV assessment in climate change at a specific location in Vietnam as far as we know. Most of the previous studies in Vietnam related to this issue only describe, or generalize based on general statistics for the whole country or a large region, without in-depth analysis. On the other hand, the authors used the modified index method to assess AV to climate change. The original index method was adjusted by the author to suit the object and context in the study area. Accordingly, the level of AV was based on a combination of the physical and socio-economic, which

included indicators related exposure, sensitivity and adaptation capacity. In this way, the extent of agricultural vulnerability to climate change can be more comprehensively and objectively analyzed, and the shortcomings of the previous studies can be eliminated.

Discussions about the exposures component pointed out that extreme weather events are more important than average annuals or seasonal changes, with the duration and intensity of events being considered particularly important. These results suggested that further investigation is needed on the missing extreme weather variables and appropriate time and intensity thresholds for the construction of agriculture-related indicators of extreme weather events. Farmers and their activities are considered as important indicators in assessing AV to climate change. Farming methods, rural infrastructure, policies related to agricultural development and response to climate change etc., are missing data that need to be supplemented to assess S and AC in a more comprehensive and complete way.

Climate change is a global issue, although climate change vulnerabilities will be quantified in a particular case. However, climate change research on agriculture still has a difficulty in explaining transnational climate impacts; that is, how global flows affect regional vulnerability and adaptive capacity. For example, how crop failure affects global food security and social security. Therefore, transnational impacts in risk and vulnerability assessments need to be considered.

## 6. Conclusions

The results of this study contribute to the analysis of indicators assessing vulnerability to climate change. The results of the study identified several important challenges in applying existing indicators and assessment methods to agricultural vulnerability in Ha Tinh province. The agricultural vulnerability needs to be comprehensively assessed on several aspects. From the present study, it was found that the combination of three components i.e., exposure, sensitivity, and adaptive capacity according to IPCC to assess agricultural vulnerability is quite reliable. Climate change has a profound effect on agriculture in Ha Tinh province, causing negative impacts on the economy, society, and the environment.

The results also show that adaptive capacity is inversely proportional to vulnerability, greatly influencing other components. The lowest levels of agricultural vulnerability were found in Ha Tinh city and Hong Linh town. This indicates that the more adaptive capacity areas are, the less vulnerable they are and vice versa. Therefore, improving adaptive capacity such as raising people's education level, improving infrastructure, and increasing the budget for climate change response are necessary solutions to minimizing the damage caused by climate change to the economy—society in general and the agricultural sector in particular.

In Ha Tinh province, the coastal districts are more vulnerable to agriculture than the others as these are areas directly affected by natural disasters such as storms, floods, and rising sea levels. In addition, low education levels and poor infrastructure in mountainous areas also make the vulnerability to agriculture higher.

From the actual results of AV studies at the national and regional levels in Vietnam [58–60], the local level research adds depth to the complex reality of vulnerability, thereby offering specific solutions suitable to the situation of the locality.

The regional quantitative assessments method identified the major drivers of agricultural vulnerability. As a result, the AV zoning map produced using GIS located the high-risk area in space. Although this study has its limitations, it contributes to the debate on how to address the threat of climate change. The results of the vulnerability assessment have been useful in generating planning measures to limit the damage and negative impacts from climate change to the agricultural sector in particular and the socio-economic of the study area in general.

**Author Contributions:** L.V.H. and D.T.L. came up with research idea through discussion. Investigation and data collection were performed by L.V.H. Data processing and the first manuscript were conducted by D.T.L., P.A.T., N.T.H.N., T.T.Q.H. and B.T.H.M. commented and approved the manuscript. All authors have read and agreed to the published version of the manuscript.

**Funding:** This research was funded by Vietnam Academy of Science and Technology, grant number UQĐTCB.02/22-23.

**Informed Consent Statement:** Not applicable.

**Data Availability Statement:** The data that support the findings of this study are available from the corresponding author upon reasonable request.

**Acknowledgments:** This research was supported by the Vietnam Academy of Science and Technology (VAST) with project code UQĐTCB.02/22-23.

**Conflicts of Interest:** The authors declare no conflict of interest.

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
