# Peer review of "An Assessment of Agricultural Vulnerability in the Context of Global Climate Change: A Case Study in Ha Tinh Province, Vietnam"

_sustainability, doi:10.3390/su14031282_

Round 1
Reviewer 1 Report
This manuscript deals with the agricultural vulnerability assessment to climate change. It is considered well organized for the journal format, however there is in need of some changes.
L 53: recommend to change the currency to euro or dollar, because is an international journal and for other authors is more simple to understand it.
L 92: "there" should be there.
L 91, 92, 93: rewrite the sentence is not clear.
L 138: from what type of Hydro-meteorological center was the data collected.
L 139, 140: from what type of meteorological stations are from which the data is obtained.
L 148 (Table 1.): in the Table there are some citations, which I did not found in the References section, also the citation type is not correct, please check in the journal template. Please check the whole Table 1.
L 168, 181, 183, 185, 197, 212, 213, 247, 290, 293: the citations of the Figures are not correct, check the journal template. Please check the whole manuscript.
L 186, 270, 290, 311, 320: “Figure 5 shows the distribution of the number of very hot days per year in the study area.” This sentence is not necessary, I recommend to place it in the figure title this. Also please check the whole manuscript, because there are similar sentences.
L 252-253: Please rewrite this sentence, is not clear what you want to say.
L 308: Table 3 and Figure 11 is the same, is a repetition, delete one.
L 348: Please rewrite this sentence.
L 352: Please correct “Moreover.”
Finally, the Discussion sections is poorly written. Authors should discuss their findings and compare them with previous findings and studies.
Reviewer 2 Report
Integrating indicators in agricultural vulnerability assessment to climate change in Ha Tinh province, Vietnam
The content of this manuscript is interesting; however, the presentation is weak. Therefore, I have to reject this manuscript and request authors to re-structure the content in a presentable way.
For example
“As a result, areas with very high AV levels accounted for 22% of the total area.”
Please try to understand the sentence. This may be understood by the authors, because they know what they have done; however, not by the readers. Therefore, the manuscript has to be re-structured.
Title
Revise the title for better understanding and clarity. The title at present looks uncleared.
Abstract- Should be re-written using correct way of academic writing.
Introduction – The section has a good count of references. However, authors should be focused on the research gap. And how it has been addressed by you. This is important.
Study area – This is quite acceptable. Please check for language issues.
Materials and methods – This is a mix of intro too. Table 1 should be explained well. What is the resolution of climate data?
“Then they are processed and interpolated using GIS software.” What is the version of GIS, be comprehensive on what you are writing.
Re-write the equations properly using an invisible table.
Results – This has to go with the discussion. If not it is not comprehensive. Combine them.
Discussion and conclusion – What are your solid conclusions? Please re-write!
References- Good number of references were cited in the manuscript.
Reviewer 3 Report
This study evaluated agricultural vulnerability (AV) to climate change in Ha Tinh province, Vietnam. Three components (Exposure, Sensitivity, and Adaptive capacity) were included in the AV assessment. The aim is clear and can provide useful information to decision-makers. However, the manuscript did not structure well, particularly in Results and Discussion. Introduction should be polished and Discussion needs to be revised to improve the clarity. Some grammar errors and word choices should be checked. For example, in L93-99, objectives and MM should be written in the past tense not present tense. Below are some specific comments on the manuscript:
L103-109: I recommend moving this part to MM. Furthermore, introduction of agriculture production would help readers understand the importance of this study.
L171: The authors somehow mixed Results and Discussion in this section. Therefore, the true Discussion section (L330) became too short and did not discuss sufficiently. Some sentences should be moved or combined in MM, such as L174-177 and L204-211, since they have been mentioned in MM.
Fig. 3-5: Why are there only data from two weather stations? When the linear regression was performed, please provide values of R square and P-value of the slope.
Table 3 and Figure 11: Are they the same data set?
Round 2
Reviewer 1 Report
Dear authors,
How suggested before the Discussion section must be improved. Authors should compare and discuss their findings with previous studies.
Author Response
Reviewer: How suggested before the Discussion section must be improved. Authors should compare and discuss their findings with previous studies.
Response
Dear Reviewer 1,
On behalf of the authors, I would like to thank you for taking the time to review our first revised manuscript. After your initial comments, we tried to revise the manuscript as good as possible, but it still has many shortcomings. In this second revised manuscript, we continue to review and correct errors in grammar, expression, and content. In the discussion section, we added a short paragraph describing the contributions of this study compared to previous research. All revisions made to this manuscript were marked up using the “Track Changes” function, which help you view easily. Please check details in attached file. Thank you again!
Best regard
Reviewer 2 Report
Authors have significantly improved the quality of their manuscript; therefore, I think, the manuscript can be moved to a decision which is favour for the authors.
Author Response
Reviewer: Authors have significantly improved the quality of their manuscript; therefore, I think, the manuscript can be moved to a decision which is favour for the authors
Dear Reviewer 2,
On behalf of the authors, I would like to thank you for taking the time to review our first revised manuscript. After your initial comments, we tried to revise the manuscript as good as possible, but it still has many shortcomings. In this second revised manuscript, we continue to review and correct errors in grammar, expression, and content. In the discussion section, we added a short paragraph describing the contributions of this study compared to previous research. All revisions made to this manuscript were marked up using the “Track Changes” function, which helps you view easily. Please check the details in the attached file. Thank you again!
Best regard
Reviewer 3 Report
The manuscript has been improved, but some minor English check is required.
Author Response
Reviewer: The manuscript has been improved, but some minor English check is required.
Response
Dear Reviewer 3,
On behalf of the authors, I would like to thank you for taking the time to review our first revised manuscript. After your initial comments, we tried to revise the manuscript as good as possible, but it still has many shortcomings. In this second revised manuscript, we continue to review and correct errors in grammar, expression, and content. In the discussion section, we added a short paragraph describing the contributions of this study compared to previous research. All revisions made to this manuscript were marked up using the “Track Changes” function, which helps you view easily. Please check the details in the attached file. Thank you again!
Best regard